# Mis–Dis Information in COVID-19 Health Crisis: A Narrative Review

**DOI:** 10.3390/ijerph19095321

**Published:** 2022-04-27

**Authors:** Vicente Javier Clemente-Suárez, Eduardo Navarro-Jiménez, Juan Antonio Simón-Sanjurjo, Ana Isabel Beltran-Velasco, Carmen Cecilia Laborde-Cárdenas, Juan Camilo Benitez-Agudelo, Álvaro Bustamante-Sánchez, José Francisco Tornero-Aguilera

**Affiliations:** 1Faculty of Sports Sciences, Universidad Europea de Madrid, Tajo Street, s/n, 28670 Madrid, Spain; vctxente@yahoo.es (V.J.C.-S.); juanantonio.simon@upm.es (J.A.S.-S.); josefrancisco.tornero@universidadeuropea.es (J.F.T.-A.); 2Grupo de Investigación en Cultura, Educación y Sociedad, Universidad de la Costa, Barranquilla 080002, Colombia; 3Grupo de Investigacion en Microbiologia y Biotecnologia (IMB), Universidad Libre, Barranquilla 080002, Colombia; eduardoi.navarroj@unilibre.edu.co; 4Psychology Department, Universidad Antonio de Nebrija, 28240 Madrid, Spain; abeltranv@nebrija.es; 5Vicerrectoría de Investigación e Innovación, Universidad Simón Bolívar, Barranquilla 080005, Colombia; cacelaca6@gmail.com; 6Facultad de Ciencias Sociales y Humanas, Universidad de la Costa, Barranquilla 080002, Colombia; jbenitez@cuc.edu.co

**Keywords:** misinformation, disinformation, COVID-19, pandemic, health crisis, social media

## Abstract

Background: In this narrative review, we address the COVID-19 pandemic mis–dis information crisis in which healthcare systems have been pushed to their limits, with collapses occurring worldwide. The context of uncertainty has resulted in skepticism, confusion, and general malaise among the population. Informing the public has been one of the major challenges during this pandemic. Misinformation is defined as false information shared by people who have no intention of misleading others. Disinformation is defined as false information deliberately created and disseminated with malicious intentions. Objective: To reach a consensus and critical review about mis–dis information in COVID-19 crisis. Methods: A database search was conducted in PsychINFO, MedLine (Pubmed), Cochrane (Wiley), Embase and CinAhl. Databases used the MeSH-compliant keywords of COVID-19, 2019-nCoV, Coronavirus 2019, SARS-CoV-2, misinformation, disinformation, information, vaccines, vaccination, origin, target, spread, communication. Results: Both misinformation and disinformation can affect the population’s confidence in vaccines (development, safety, and efficacy of vaccines, as well as denial of the severity of SARS-CoV infection). Institutions should take into account that a great part of the success of the intervention to combat a pandemic has a relationship with the power to stop the misinformation and disinformation processes. The response should be well-structured and addressed from different key points: central level and community level, with official and centralized communication channels. The approach should be multifactorial and enhanced by the collaboration of social media companies to stop misleading information, and trustworthy people both working or not working in the health care systems to boost the power of the message. Conclusions: The response should be well-structured and addressed from different key points: central level and community level, with official and clearly centralized communication channels. The approach should be multifactorial and enhanced from the collaboration of social media companies to stop misleading information, and trustworthy people both working and not working in the health care systems to boost the power of a message based on scientific evidence.

## 1. Background

During the 21st century, the possibility of a new pandemic was already a risk the World Health Organization (WHO) was aware of. In 2011, the WHO already presented a worldwide initiative, “Pandemic Influenza preparedness Framework”, designed to stop or delay a pandemic situation in its initial stage and therefore prevent it from spreading worldwide [1]. Despite governments and world organizations’ efforts, nobody was prepared for the outbreak of a contagious disease such as COVID-19. On the 11th of March of 2020, the WHO officially characterized COVID-19 as a pandemic with more than 118,000 cases in 114 countries and 4291 deaths, while thousands were fighting for their lives [2,3]. Currently, as of on 14th of March 2022, cases have ascended to 458 million in over 190 countries and 6.04 million deaths (World Health Organization, 2022).

During the COVID-19 pandemic, healthcare systems have been pushed to their limits, with collapses occurring worldwide. The pressure imposed by COVID-19 cases on healthcare systems nearly collapsed them in the most affected areas during the first wave of the pandemic. It occurred in Europe’s already debilitated health systems, resulting from years of fragmentation and decades of finance cuts, privatization, and deprivation of human and technical resources [4]. In Italy, for example, the National Healthcare Service suffered a cut of more than EUR 37 billion over the period 2010–2019, while also suffering the progressive privatization of healthcare services [5]. Lombardy was the most affected region in Italy, with 1006 patients requiring advanced respiratory support, where Lombardy had only a capacity of 724 intensive care beds [6]. Italian Civil Protection undertook a fast-track public procurement to secure 3800 respiratory ventilators, 30 million protective masks, and 67,000 tests [7]. To avert the shortage of health workers, the Italian Government authorized regions to recruit 20,000 health workers, allocating EUR 660 million for the purpose [8]. Similar situations happened in other countries: in Spain, the central government adopted a series of financial measures to support the health system and protect businesses. It had allocated EUR 2800 million to all regions for health services and created a new fund with EUR 1000 million for priority health interventions [9]. Health facilities in the most affected regions struggled with inadequate intensive care capacity and an insufficient number of ventilators. Both Catalonia and Madrid cancelled non-emergency surgery and cleared beds where possible. Telephone helplines had long delays or simply collapsed. Regions were allowed to take over the management of private health services, while military installations were used for public health purposes [10]. Medicine and equipment shortages encouraged profiteering, with private laboratories charging exorbitant amounts for tests [11]. In response, the central government of Spain centralized purchasing and introduced price controls on medicines requiring companies producing relevant equipment to inform the central government of their stocks within 48 h [12]. To avert the shortage of health workers, insufficient measures were suggested such as cancelling holidays or bringing retired nurses and doctors back into the health service, resulting in the exhaustion of healthcare workers. The shortage of healthcare workers became exacerbated by the quarantining of a growing number of health workers exposed to infected patients. Thus, the Spanish government started to hire graduate medical students without specialization and nursing students, as well as extending contracts of medical residents [13].

Yet, COVID-19 has had a global impact, not only medically, affecting environmental sustainability and economic development [14]. Social restrictions have led to a decline in energy demand and industrial output. Among these measures, social distancing has led to the closure of schools, universities, libraries, centers for older people, and sporting venues, and even restricting all movement in some of the most affected areas [15]. Shutting down borders and all types of travel between countries and the mandatory or voluntary wearing of face masks depending on the country or region [16]. This translated to a reduction in coal and oil consumption, which led to a 25% decline in carbon dioxide emissions, corresponding to a 6% reduction in global emissions [17], with a 70% reduction in nitrogen oxide emission [18]. Furthermore, the shutdown of the tourism industry led to an improvement in biodiversity and enhanced the regenerative capacity of the marine habitat (fishing ground) and forest reserve. Furthermore, the shutdown of power plants and commercial and manufacturing activities led to a reduction in ambient noise levels [19]. However, a temporary reduction in atmospheric emissions and environmental degradation in the period of the global pandemic does not translate into total environmental sustainability [20]. Indeed, there are countries that have significantly increased their pollution levels [21], as in the case of China, with an increase of 240 metric tons of medical waste [22].

Yet, the stoppage that occurred during the period of 2020 and continued uninterruptedly during 2021 has affected global economic development, resulting in several fiscal measures, monetary policies, and private sector economic burden sharing across countries [23]. The slowing down of the Chinese economy with interruptions in production altered the functioning of global supply chains. Thus, companies around the world with ties to China experienced a reduction in production, transport limitations, and border closures, slowing the global economy. This triggered panic among consumers and firms, while global financial markets were also responsive to the changes, and global stock indices plunged [24].

This situation has been challenging companies in various sectors of activity, leading to the digitization of processes and activities [25]. During quarantine, people had to adapt to a teleworking model and learn to share a reduced space with their relatives. Domestic tasks and office work coexisted at home. Having a big impact on people’s lives and existing a greater risk than before of having their income reduced or losing their jobs [25], which according to authors, approximately 60 million Europeans were at risk. [26]. Likewise, the distance between social classes and differences began to increase during the pandemic [27], where the self-employed and small businesses in sectors such as restaurants, transport, tourism and leisure were the most affected [28]. 

This extended to a significant increase in psychological and psychiatric disturbances that can prevail for months or years [29,30]. Additionally, one of the most notorious is the reduction in levels of physical activity; according to the authors, COVID-19 has aggravated the obesity pandemic by the creation of an unprecedented obesogenic environment during quarantine [31] with an increase in sedentary behavior: irregular sleep patterns and extensively prolonged screen times because of online classes, online lectures, or online work and increased leisure screen time [32,33]. This has led to weight gain and reduced cardiorespiratory fitness levels [34]. All these are factors that feed the infectious and aggravating vicious cycle of the disease and symptoms of COVID-19 [35].

The pandemic has changed life as we know it, on behaviors, quality, and perception of life. All this added to the context of uncertainty has resulted in skepticism, confusion, and general malaise among the population.

In the actual COVID-19 pandemic, we have witnessed a high amount of fake news and false, partisan, and disinformation, with different origins, different objectives, and, as a target audience, the world population. Then, to analyze the mis–dis information during the COVID-19 health crisis and in the vaccination process we conducted the present narrative review. In the present research, we will discuss the origins of COVID-19 mis–dis information, identify catalysts for the resultant infodemic, and suggest methods for mitigating false information online. 

The aim of the present narrative review was to analyze the mis information and dis information during the COVID-19 Pandemic crisis and in the COVID-19 vaccination process. For this goal, the origin of mis–dis information, target, spread and main communication channels used will be analyzed and discussed.

## 2. Materials and Methods

To reach the study aim, a consensus and critical review was conducted, analyzing primary sources such as academic research and secondary sources such as databases, web pages, and bibliographic indexes, following procedures of previous critical narrative reviews [36,37,38,39]. We employed in PsychINFO, MedLine (Pubmed), Cochrane (Wiley), Embase, and CinAhl. Databases the MeSH-compliant keywords of COVID-19, 2019-nCoV, Coronavirus 2019, SARS-CoV-2, misinformation, disinformation, information, vaccines, vaccination, origin, target, spread, and communication. We used manuscripts published from 1 January 2020 to 20 November 2021, although previous studies were included to explain some information in several points of the review. We used the following exclusion criteria in line with previous reviews [38,40,41]: i. research outside the time period analyzed, ii. presented topics out of the review scope, iii. unpublished studies, books, conference proceedings, abstracts, and PhD dissertations. We used all the studies that met the scientific methodological standards and had implications with any of the subsections of the present review. The treatment of the information was performed by all the authors of the review, and finally, the articles selected were discussed to write the present review. 

## 3. Results and Discussion

The aim of the present narrative review was to analyze misinformation and disinformation during the COVID-19 pandemic crisis and in the COVID-19 vaccination process. Our main findings were related to bear in mind the stress and uncertainty of the citizens, the lack of resources to fight against a pandemic, and the need to fight misinformation and disinformation with a well-structured, prepared, and advanced strategies. Transparency is a key factor that institutions should centralize through official communication channels. A multifactorial approach is needed to stop misleading information.

### 3.1. Mis–Dis Information in COVID-19 Crisis

Misinformation is defined as false information shared by people who have no intention of misleading others. Disinformation is defined as false information deliberately created and disseminated with malicious intentions. Both misinformation and disinformation can affect the population’s confidence in the cost benefits of the vaccines that are currently being applied in the countries. Most of the misinformation and disinformation that has circulated about COVID-19 vaccines has focused on the development, safety, and efficacy of vaccines, as well as denial of the severity of SARS-CoV-2 infection [42]. Both terms have been identified even before SARS-CoV-2 was identified in 2020, on a large number of health topics whose common pattern is the online dissemination of potentially harmful information [43]. The amount of false information that circulates on social networks is so great that an individual strategy by countries is necessary, which allows us to effectively combat misinformation and misinformation related to the health-disease process, both for COVID-19 and for other diseases.

#### 3.1.1. Origin of Mis–Dis Information

In the last few years, there has been a significant increase in campaigns related to misinformation, disinformation, rumors, fake news, and different conspiracy theories. To a large extent, this phenomenon has undergone a process of acceleration due to the Internet capacities, modern information technologies, digitization processes, and the impact of social networks [44,45]. The COVID-19 pandemic triggered a wide consumption of information and, in parallel, a clear process of “infodemic”, according to the World Health Organisation (WHO), took place. An “infodemic” is an overabundance of information—in some cases accurate and in others not—that is disseminated during an epidemic and that spreads among humans in an epidemic-like manner through digital and physical information systems [46].

The first cases of viral pneumonia of unknown cause were reported in a statement by Wuhan (China) officials on 31 December 2019. The WHO declared that COVID-19 had reached pandemic status on 11 March, meaning that the epidemic had spread globally, crossing international borders [46]. At that time, European countries such as Italy and Spain became the epicenter of the crisis before its rapid spread to other countries such as England, the United States, India, and Brazil. Immediately, an exponential increase in the spread of misinformation, hoaxes, and deliberate lies related to the COVID-19 pandemic could be seen spreading through the media and social networks [47,48,49].

Many fake or misleading stories began to be fabricated and shared among thousands of people without any control of their veracity or quality. Much of this information was based on conspiracy theories related to how the virus originated, its cause, its treatment, and its mechanism of spread and contagion. The spread of mis–dis information caused many people’s behavior to change, leading them to take greater risks and jeopardizing the sustainability of the healthcare system. In the following weeks, social media, WhatsApp chats, and private mailboxes were overwhelmed by an avalanche of information, many of which were correct and authentic, but most were unusable. In some situations, this confusion also came from the political and scientific authorities themselves, as was the case with the controversy that arose over the use of anti-inflammatory drugs and their relationship with COVID [50].

Just after COVID-19 was declared a Public Health Emergency of International Concern, the WHO launched a platform called the WHO Information Network for Epidemics (EPI-WIN), which aimed to reduce misinformation by facilitating access to timely and accurate information about the pandemic from trusted sources [3]. Sylvie Briand, director of Infectious Hazards Management at the WHO’s Health Emergencies Programme, stated that, in this type of situation, it was common to be accompanied by “a kind of tsunami of information”, but that false rumors, misinformation, disinformation, hoaxes, etc., were also always encountered. However, Briand pointed out that the difference with phenomena that have been occurring in our societies since the Middle Ages lies in the incomparable impact of the new social networks that cause a greater speed and expansion of misinformation [51].

The European Commission also tried to stop the effects of mis–dis information campaigns that grew with the onset of the pandemic. In April 2020, Věra Jourová, Vice President for Values and Transparency in the European Commission, harshly criticized large U.S. technology companies such as Google and Facebook “for making money off coronavirus-related fake news instead of putting in more efforts to stop the deluge” [52]. Meanwhile, companies specializing in the fight against disinformation, such as Blackbird.AI, warned this month that “COVID-19 is the Olympics of disinformation”. Blackbird’s COVID-19 Disinformation Report analyzed 49,755,722 tweets from 13,203,289 unique users on topics related to COVID-19 for the period of 27 February–12 March 2020. Their results showed that 18,880,396 million tweets had manipulated content [53]. Reporters Without Borders (RWB) reported that between 20 January and 10 February, 2 million of the messages posted on Twitter, 7% of the total, were spreading conspiracy theories about coronavirus [54].

The United Nations Educational Scientific and Cultural Organization (UNESCO) identified some of the common formats in which misinformation is disseminated through text, images, videos, and audio. Specifically, it mentioned the creation of fake websites and fabricated identities through which false news related to COVID-19 is reported. A second format refers to the use of fabricated, altered, or decontextualized videos and images in order to create doubts and mistrust among the population. In addition, emotive narrative constructs and memes become viral through messaging apps—especially WhatsApp—making it very difficult to identify and combat this type of misinformation. Finally, disinformation infiltrators and orchestrated campaigns seek to activate social conflicts and confrontations by promoting geopolitical interests and the most nationalistic sectors of some countries.

Undoubtedly, new technologies have enabled our society to cope better with this type of pandemic, helping specialists to connect with each other in real time or allowing governments to inform citizens directly about public health issues. Nevertheless, it has been proven that mass connectivity has also allowed misinformation to spread virally, endangering the population, undermining democratic principles, and damaging trust in scientists [45,48]. 

#### 3.1.2. Target of Mis–Dis Information

Mis–Dis information related to COVID-19 has been instrumentalized for a variety of political, racist, and sexist purposes, among many others, which have sought to polarize public opinion and foment hatred at a time when global solidarity and cooperation are needed to confront this epidemic. However, who are those responsible for spreading this mis–dis information and making it viral, and who are these mis–dis information campaigns aimed at? Regarding the first aspect, there are a wide variety of actors who contribute to spreading this misinformation. In an article, the BBC attempts to classify the different actors who start and spread viral COVID-19 misinformation: conspiracy theorists, politicians, and celebrities [55]. 

According to a Reuters Institute research survey, high-level politicians, celebrities, or other prominent public figures produced or spread only 20% of the misinformation in analyzed in their research, but that misinformation attracted a large majority of all social media engagements. Based on the same survey, 87% of online pandemic disinformation disseminated between January and March 2020 used repurposed content, supplanted genuine sources, or updated long-standing conspiracy theories [56]. US President Donald Trump suggested that disinfectants could kill the virus [57], and Madagascar president Andry Rajoelina showed his support for an unproven herbal tea to cure COVID-19 [48]. These are some examples of statements that have helped spread mis–dis information, jeopardizing the safety of our societies and reducing trust in international agencies such as the WHO.

Some research that has analyzed the misinformation generated by the media between 1 January and 26 May 2020 has highlighted that conspiracy theories spread by antivaccine groups, 5G opponents, or political extremists contributed much less to the overall volume of misinformation than other more powerful actors such as the U.S. president [58].

This type of news coverage, mis–dis information, and fake news have led to an increase in racism and exclusion of certain social groups. In many countries, racist behavior and discriminatory treatment in daily life and in the media against the Chinese population were noted due to the origin of the first COVID cases [59]. At the same time, extremist groups have also used online platforms to hold vulnerable communities such as migrants, refugees, Jews, or Muslims responsible for the spread of COVID-19 and target them for violent behavior. Groups with limited access to healthcare and working in precarious jobs such as people of color and migrants have been more directly affected by coronavirus. In some cases, these groups find it more difficult to achieve self-isolation, increasing the likelihood of spreading the virus; in other situations, mainly undocumented immigrants avoid hospitals for fear of being identified and reported, which ends up causing them to suffer a stronger impact of this disease [60]. Political leaders such as Matteo Salvini, former Deputy Prime Minister of Italy, attacked the president of the Italian government for not closing the borders to African immigrants by linking them to the spread of COVID-19 in this country [60]. 

According to Europol’s reports, “some state and state-backed actors seek to exploit the public health crisis to advance geopolitical interests, often by directly challenging the credibility of the European Union (EU) and its partners” [61]. Lea Gabrielle, Special Envoy and Coordinator of the Global Engagement Centre at the US State Department, stated in March 2020 that countries such as Russia, China, or Iran were implementing misinformation and fake news campaigns in order to disrupt the world order and promote their political and economic interests [62]. Nevertheless, some authors have indicated that the content of these specific state-sponsored sources represents only 0.77% of the data set they analyzed, with credible sources being much more numerous [63]. For the European Union these types of misinformation campaigns driven by some countries seek to undermine human rights and many of the fundamental principles of democratic systems. 

Another objective of the COVID-19 infodemic has been to damage public confidence in science and endanger public health. From the first moment the pandemic began, multiple theories about the origin of the virus took hold on the Internet, all of which had the common theme that the virus had been created in a laboratory by a rogue government with an agenda. This misinformation was created on social media accounts and websites without credible evidence to support their claims and accumulated millions of visits. Scientists from several countries have analyzed the COVID-19 genome and came to the decisive conclusion that the virus originated in nature from an animal source [64]. Subsequently, other conspiracy theories about the origin of the disease mentioned that it was 5G networks and not coronavirus that made people sick [58], that Bill Gates had created this virus, or that it was chemical weapons manufacturers [65]. These mis–dis information campaigns have underscored the importance of producing and disseminating credible versus unreliable information to the public and the important role played in this process by the medical community [66,67].

Finally, mis–dis information campaigns have affected the credibility of journalists by casting doubt on the information they disseminated through the traditional media. When journalists attacked these misinformation campaigns, they were attacked with more misinformation [68]. Similarly, in relation to the economic impact of COVID-19 and the social isolation measures that were implemented at the height of the pandemic, mis–dis information was disseminated in many countries that sought to discredit these types of measures by suggesting that social isolation is not economically justified. Cybercriminals also found in hoaxes, fake news, and mis–dis information new opportunities to generate frauds that sought to steal people’s private data or to enrich themselves through the sale of sanitary products and masks [69].

#### 3.1.3. Spread of Mis–Dis Information

Informing the public has been one of the major challenges during the COVID-19 pandemic. Since the beginning, institutions and authorities have been providing information, including through social media, while fighting against fake news [5]. However, the broad power and spectrum of social networks and information residing on the internet added uncertainty and fear to the exceptional situation, surprising governments and societies, resulting in: “The First True Social-Media Infodemic” [70].

Misinformation or disinformation has been around for centuries, ever since the free press existed. It has shown a greater capacity to spread faster and farther than accurate information, having a truly negative impact in the world, such as amplifying controversies about vaccines [71]. However, there is a conceptual discrepancy between these two terms. Misinformation would be defined as false information spread without malicious intent, while disinformation is used by governments, militaries, organizations, and individuals to intentionally mislead or manipulate the public. Differences between these two terms underlies the intention behind the spreading of false/fake information. 

Nowadays, access to information is so easy and fast that the world is exposed to a “misinformation pandemic”. Undoubtedly, the consolidation of technologies and the internet has given the world an abundance of news and information, where keeping informed truthfully is not easy, especially due to the politicization of the media [72,73,74]. In this line, the COVID-19 pandemic has exposed health inequities and disparities, bringing into light the phenomenon of health misinformation. Etiologically, it would be defined as fake facts or falsely claimed health-related information that contradicts the scientific community and medical consensus. A large group of the population uses information on the Internet to obtain information in clinical and medical terms about health. Two-thirds of U.S. adults have sought information about their health online in the past year [75], and recent studies suggest that it has grown with COVID-19 to up to 75% of the U.S. population [76], translating this into approximately more than a billion searches in search engines such as Google per day. 

This is a double-edged sword, since digital channels are valuable mechanisms to provide accurate health information and guidance to the public; however, they have become gateways for the rapid spread of health misinformation and disinformation [77]. Indeed, 80% of people who search for health information online do not know how to distinguish between misinformation and fact [78]. Epidemiologists have compared the spread of online information to the transmission of infection, making an educated public key to reducing the spread of disease [79]. Thus, health misinformation urgently requires greater action from those working in public health research and practice [80]. 

During this stage, the COVID-19 pandemic period, where the number of searches and misinformation especially increases, authors have called this “The First True Social-Media Infodemic” [70]. It was not until mid-February when the WHO decided to declare among the “COVID-19 health pandemic” the situation of an “infodemic” [81]. We must understand that the social context in which there is a dwindling trust in politicians, science in general, rampant conspiracy theories, with the added effect of the isolation of many people during the pandemic, has led to this perfect breeding ground for profiling antimaskers and COVID-19 denialists [82]. Among the different conspiracy theories and fake news, those that had the greatest impact were: the virus is a secret attempt by the global elite to reduce overpopulation; the virus is a bioweapon of the Chinese state to control the world; the virus is a scheme by greedy “Big Pharma” companies to make money off vaccines; eating garlic, drinking hot water, not eating ice cream, or wearing salt-coated face masks will keep the virus at bay; drinking bleach, chlorine dioxide, colloidal silver, or your own urine can help kill the virus [70].

In this line, social media such as WhatsApp, Twitter, YouTube, and Facebook have started to fact check, label, and limit the spread of misleading information, including the removal of fake news [56,83]. However, these actions can be considered a “quick-patch”, and many times, predictive algorithms aimed at limiting the information by large platforms can be perceived as an attack on individual liberties and freedom of expression. Therefore, this patch can cause even more damage among the general public, leading to greater uncertainty and belief in informational bias. Since digital communication channels are here to stay, continued efforts to combat misinformation and disinformation are critical. Yet, their key resides in building trust in communities, publishing and sharing verified information, improving media and health literacy, and ultimately, educating the citizens [77].

The consequences of this infodemic are directly related to the process of assimilation of the disease by populations, encouraging them to make wrong decisions, such as the cases of bleach intake (cleaning and disinfection product) in the United States, where rumors commented that it could help kill the COVID-19 virus [84]. In South Korea, attendees of a church became infected with COVID-19 after receiving a saltwater spray in their mouths by their leaders, believing that this solution helped to reduce the spread of the disease; the instrument used for the aspersion was the vehicle that allowed the contagion between people [85]. In Iran, around 300 people died from ingesting methanol as a treatment for the COVID-19 disease [86].

It is also important to consider that, given the appearance of new large-scale events such as a pandemic, the uncertainty generated by identifying what is true or false, because the scientific evidence is limited and changing, will influence the behavior and beliefs of the community in general [87]. However, Hameleers et al. identified that people with a perception of misinformation could be motivated to seek information and stimulate benevolent behavior [88].

#### 3.1.4. Main Communication Channels Used

Facing emergency events such as a pandemic, several questions arise that need to be answered immediately and with truthful information. Currently, the media such as the press and local radio have been relegated to the background by their audience, giving greater importance to social networks such as Facebook, Instagram, Twitter, WhatsApp, and YouTube, which play an important role when spreading a message, whether it is real or not [89]. 

In the COVID-19 pandemic, it has been observed how these mass media have an impact on how the world and daily life are perceived, providing reasons for misinformation related to the disease. These social networks have been the source of conspiracy theories about the creation of the virus as a means of creating a biological war against China to suppress its economic growth [89], as well as the beginning of xenophobic attacks against the Chinese population, such as the trend on Twitter under the hashtag #ChineseDon’tComeToJapan when spreading erroneous information that some passengers residing in Wuhan who presented symptoms of fever violated the quarantine at the Airport Kansai International [59,72].

Likewise, these giants of social networks have made an effort to reduce the false information that is disseminated through their platforms; in the case of Facebook, it has created the Information Center on Coronavirus, in which they publish updated and secure information on the progress of the pandemic [90]. Additionally, Google created an alert in which searches for information received by the public come from the official pages and networks of the World Health Organization and other authorized sources [72].

In this sense, the media is positioned as a fundamental part of the management of any health crisis, since despite the restrictions generated through filters and search generators of social networks, it is not enough to mitigate the dissemination of fake news [90]. eHealth literacy, which is a set of skills based on media literacy that provides people with the ability to navigate online resources and assess effectively the sources of information to make a well-informed opinion, is a key factor to combat misinformation and disinformation threats [91]. 

#### 3.1.5. Consequences

Since the global decision of confinement to prevent the rapid spread of the virus, social interactions have been drastically reduced [92]. For this reason, social networks and information transmitted through new technology channels became essential to connect people and keep them informed. At a time when there was a great lack of knowledge about the virus, the news was the only way to try to understand what was happening and make sense of the extreme measures that were being implemented worldwide [93]. Thus, data were provided on aspects such as the number of infected people, the number of deaths, the impossibility of providing health care to the sick, and confusion and uncertainty about the near future, among other fears in the general population.

During the pandemic, this information reached the general population on a massive scale, transmitted by communication professionals, health professionals, politicians, scientific disseminators, virologists, and journalists, who gave different visions of the virus [94]. This panorama of public uncertainty and the facility to transmit information to people confined to their homes made it easier for many media reports to spread unverified news, news that had not been adequately verified by professionals dedicated to public health, and other news without an accurate and properly tested scientific basis.

When research on COVID-19 began, governments, pharmaceutical companies and the health industry invested a lot of resources in trying to understand how the virus works and to reduce the spread and number of people affected [95,96]. On this dizzying and innovative path in the current times, the lack of rigorous information and the partial information that reached the population facilitated the emergence of groups of people who began to doubt the veracity of the data and the existence of the pandemic.

The consequences of a constant inundation of scientifically unreliable information have led to the emergence of another pandemic, a global infection of data that has made the public not trust what is transmitted by the main channels of communication [97,98]. This has facilitated increased fears about COVID-19, such as it being is possible sequelae, or the concerns over the reality about the number of people affected in the world. Ultimately, it permitted many people to create movements against the existence of the disease, supported even by health professionals and other fields of expertise [99,100]. 

This has meant that the measures adopted were not sufficiently relevant to the population, were not respected because they were not believed in, or were even expressly and manifestly disobeyed by the rest of the people, the media, and health, political, and police authorities [101]. This type of actions, which were backed up by unreliable and unverifiable news, obstructed all the measures implemented at the beginning of the spread of the virus and, to this day, its negative effect continues to be present in other relevant areas, such as vaccination [56].

In the last year, medical experts, health authorities and governments around the world have had to deal with two situations that have had a direct impact on the progress of the disease [79]. On the one hand, the presence of an unknown virus, which has spread rapidly and globally, making it necessary to take decisions to avoid endangering people’s lives. Additionally, on the other hand, challenging all the information that was made available to the population and that could endanger the integrity of the people and of the measures taken to protect the public [102].

### 3.2. Mis–Dis Information in COVID-19 Vaccination Process

Within the COVID-19 pandemic, we find another important moment and where misinformation has been an everyday element. We are talking about the vaccination process. With a series of vaccines developed in record time and with few studies of their validity, reliability, and safety, the amount of information about them has overwhelmed the ordinary citizen of the world. In this section, we are describing the origins of mis–dis information in the COVID-19 vaccination process and the aspects to implement an efficient strategy to combat mis–dis information for COVID-19 vaccination process.

#### 3.2.1. Origin of Mis–Dis Information

Since the 19th century, vaccines have had a major impact on health around the world, resulting in the eradication of diseases such as smallpox, and preventing millions of deaths a year [103]. Smallpox, the disease that accompanied humanity for at least 3000 years, was officially declared eradicated by the WHO in 1980; that is, more than half of the world’s population has not been vaccinated against smallpox, but there is no risk of an outbreak.

However, there is a long tradition of vaccine vacillation, which refers to a delay in the acceptance or rejection of vaccines, despite the availability of vaccination services [104], even before the appearance of the Internet and its services, which offer the possibility to quickly disseminate information of all kinds. However, fear of the risks of vaccination is not irrational because there are examples of vaccination-related deaths. In 1955, a contaminated batch of polio vaccines resulted in the paralysis or death of 56 children [105]. In 2014, an unusual number of events allegedly attributed to the human papillomavirus vaccine (HPV) occurred in a town in northern Colombia, which plummeted coverage and stalled national vaccination [106]. These examples and other regional events may have exacerbated vaccine concerns, especially those related to flu vaccines, as well as COVID-19 vaccines.

Many of the arguments against vaccines are based on the assumption that vaccines are ineffective and carry safety risks [105]. Sources for these arguments can have a high scientific reputation, such as a high-impact journal. One of the best-known cases is the findings by Wakefield et al. in 1998 [107]. In an article published in the Lancet, they concluded that the measles, mumps, and rubella vaccine can predispose to pervasive developmental disorders in children. Despite the small sample size (*n* = 12), the spurious design, and the speculative nature of the conclusions, this article received wide publicity (today, its retraction has more than 3000 citations registered with NCBI). Vaccination rates against measles, mumps, and rubella began to decline because parents were concerned about the risk of autism after vaccination. In 2010, the article was withdrawn from the Lancet as a fraudulent study that relies on the falsification of data [108]. However, Wakefield’s findings are still used as evidence among people warning of the (alleged and unfounded) risks of COVID-19 vaccines [109,110,111].

There is increasing literature examining how vaccine doubt is discussed on social media [112]. The question about the efficacy and safety of vaccines has gained wider support due to the lack of scientific consensus on the information and how to apply the vaccines. This has led to a greater reliance on health information generated by people with little or no scientific training, thus exposing people to misinformation or misinformation about vaccines [113]. It is often spread in emotional narratives such as blogs, Facebook accounts, Twitter, WhatsApp groups, and Reddit forums, among others, about people who have supposedly been affected by vaccines. Unlike traditional media such as the press and newscasts, published content does not need to undergo editorial selection or scientific research and can represent a more complex mix of pseudo-evidence and personal opinion [114]. Additionally, users frequently remain anonymous, allowing people to express their views without adulteration. The media are also characterized by their potential to reach large audiences and spread information very quickly [115] (Table 1). For example, in the United States, in the first 2 days of the first people receiving the Pfizer COVID-19 vaccine, antivaccine activists amplified stories of allergic reactions or even deaths caused by the vaccine, and even the outbreak of COVID-19 become its golden age [116]. Unfortunately, newspapers can be a sounding board for a few cases that discourage the population from getting vaccinated, and even expect to be vaccinated with less problematic brands. In such narratives, the risks of the vaccine may seem more immediate and tangible compared to the potential benefits of disease prevention through a vaccine [112].

#### 3.2.2. Target of Mis–Dis Information

Regarding the target of misinformation in the COVID-19 vaccine process, we can focus on two different levels of intervention, one at the central level and another more focused on the strategies followed at the community level.

##### Central Level Strategies to Address Misinformation about the COVID-19 Vaccine

Establish a multi-agency national security response effort that prioritizes the management of public health disinformation, both national and international sources, as a national security issue to prevent disinformation campaigns and educate the public on its use.Encourage active, transparent, and non-partisan intervention from the media and news media companies to identify and eliminate, control the spread of, and reduce the generators of false information.Prioritize public health risk communication at the federal, state, and local levels in public health departments and academic research by increasing research staffing, funding, and support.Increase coordination between public health experts, public information systems, and the media to increase the dissemination of accurate information through multiple channels.Promote health and digital literacy through multiple sources including schools, community organizations, social media, media, and others to help consumers choose responsible information sources and increase their awareness of misinformation tactics and approaches.Ensure multisectoral collaboration to combat public health misinformation through collective planning with social media, the media, government, security officials, public health officials, scientists, the public, and others.

###### Community Strategies to Address Misinformation about the COVID-19 Vaccine

The first step in effectively dealing with misinformation about COVID-19 vaccines is to learn more about it, including where it starts and when, why, and how it is spreading and evolving. The European Centre for Disease Prevention and Control (ECDC) issues the following recommendations to local stakeholders and health authorities to communicate accurate information about COVID-19 vaccines, respond to information gaps, and confront misinformation with evidence-based messages from trusted sources [42]:Listen to and analyze the misinformation circulating: monitor social media channels and traditional media for misinformation and create a record of misinformation to identify trends in your area. This can help to understand where, when, why, and how misinformation is spreading in the community.Interact with community members to identify and analyze perceptions, content gaps, information gaps, and misinformation.Share accurate, clear, and easy-to-find information that addresses common questions about the vaccination process and the effectiveness of vaccines. This can be achieved through official information channels (WHO). Use methods to inform people with limited or no access to the Internet, such as radio or community events. Share details, including addresses and times, about vaccination locations and events, including ages and locations with community organizations and local media. Guarantee the quantities of vaccines necessary to prevent people from making more than one trip to be vaccinated.Use trustworthy people to increase credibility and the likelihood of being seen and believed over misinformation. Some people may not trust public health professionals or visit the health department website, so it is more effective to communicate with them through the channels and sources they seek and trust to obtain health information, such as religious leaders, artists, radio hosts, athletes, influencers, or community organizations.

The United Nations International Children’s Emergency Fund (UNICEF) developed a management guide to combat misinformation to facilitate the development of strategic national action plans to quickly counter misinformation about vaccines and generate demand for vaccination through social listening. This guide is intended for healthcare professionals including national immunization plan coordinators, communication specialists, and social and behavioral change specialists. [126]. The guide is organized in three phases plus a preliminary preparation phase: listening, understanding, and participating [127] (Table 2).

#### 3.2.3. Spread of Mis–Dis Information

As we highlighted in previous sections, the WHO 2020 declared that the COVID-19 outbreak has been accompanied by a massive infodemic [128,129], referring to the overabundance of information, some accurate and some not, that occurs during an epidemic [91]. The increase in the spread of misinformation in the vaccination process of COVID-19 can largely be related to the increase in the use of social networks [128] and the amplification of the social consensus that occurs through it [130].

This has become one of the main challenges for public health, because mis–dis information can favor bad health habits and a greater spread of disease [128,130]. Exposure to false information is more common than is believed: in a survey conducted by Ofcom in the UK, 46% of the British population reported being exposed to fake news about COVID-19, and 66% of those exposed reported seeing it several times [130]. Countries such as Mexico and Spain show a greater belief in erroneous information, compared to countries such as Ireland, the United Kingdom, and the United States [130].

Factors such as distrust of science, journalists, media, government, and a conservative political ideology have been associated with beliefs in mis–dis information [131]. In addition to this, education, analytical thinking, numerical skills, and thinking styles play an important role in processing misinformation. For example, numerical skills are related to greater accuracy in judgment and decision-making [130,132]. Likewise, periods of political and economic turmoil, as well as electoral periods, have been associated with an increase in the dissemination of erroneous information [128,133,134]. The spread of misinformation has focused on older people, women, young people without college degrees, minority ethnic groups, low-income groups, and health risk groups [128,130,135]; however, the impact of misinformation on these groups is still unknown regarding COVID-19 [136].

Fake news online spreads faster than real news [91]; this topic seems to be associated with the various theories circulating in different media, such as the damage of the vaccine to the body, the 5G conspiracy theory, and the association of the vaccine with infertility, among other conspiracy theories [128,132,137]. All this affected the different antivaccine movements around the world [138] and the trust in health systems and medical staff, assisting mis–dis information, the spread of the virus, and the slowness of the vaccination process [128,133]. 

Strengthening health communication strategies including positive and safe emotions could help favor vaccination processes [139]. It is important to be guided by evidence-based strategies for proper health communication. Strategies include: (a) establishing expert group organizations to monitor and solving fake news about COVID-19, (b) equipping celebrities and politicians with scientific information about vaccines, (c) supporting vaccination through public letters and statements, and (d) having no tolerance for false and manipulated information about COVID-19 [140,141]. 

#### 3.2.4. Main Communication Channels Used

Among the main channels used to obtain information about COVID-19 vaccination process, the main were the television, radio, local newspapers, family, health authorities, the internet, national newspapers, social networks, alternative sources, medical professionals, and scientists [142,143,144]. In countries such as Germany, television, radio, and newspapers were the media outlets that used the most to obtain information on health issues, in contrast to experts who are rarely listed as sources of information [145].

In the United States, 86% of the people surveyed used traditional media (television and newspapers) to obtain information about COVID-19 vaccination. Likewise, the people who are most suspicious of the vaccination process seem to only use social networks as their only source of information [146].

Likewise, health authorities seem the most appropriate to disseminate issues about vaccination in COVID-19 because it is a widely used and reliable source. On the other hand, information on social media appears to be unreliable and useful in increasing vaccination intent. However, the approach chosen to disseminate this information must take into account each age group, because adults use more media such as newspapers; on the contrary, young people make more use of social networks and the internet [145].

In this line, all people are exposed to misinformation; however, this varies depending on their preferences and demographic data on social networks [136]. The structures of social networks, as well as their algorithms, can lead to selective exposure of disinformation in certain groups [136]. It seems that the internet has become the main source of infodemic in the world [91]. It has been found that, on the YouTube platform, 25% of the videos associated with the topic contained misleading information about coronavirus [128,130]. Likewise, media such as Twitter favor the rapid mass spread of mis–dis information [139,147,148]. However, these same platforms have also been used to provide accurate vaccination information around the world. In Italy, the Ministry of Health has made use of networks such as Facebook to mitigate the spread of mis–dis information and offer information online [149].

The importance of selecting the correct sources to inform us will have an influence on the quality of the information obtained and the susceptibility to mis–dis information [130]. Understanding the management of different communication channels and social networks themselves can help improve health communication campaigns [136]. Regarding the spread of pro-vaccine and antivaccine messages through social networks, it was found that antivaccine communications were mainly based on anecdotal stories, humor, and sarcasm. This framework of communication was often more persuasive than the use of information and participation by the pro-vaccine communications. Interestingly, both approaches used celebrity figures to spread their messages [150]. 

#### 3.2.5. Consequences

Once health authorities were able to halt the spread of the virus, governments were faced with the need to fight to contain the spread of news that tried to make the population believe that it was all a lie [151]. The combat against fake news became as important as the combat against the virus itself. This was the first pandemic to emerge in the age of communication and technology, which made it much more difficult to contain the spread of fake news [81].

Governments invested all the resources at their disposal to combat these two directions [152]. However, the creation of a vaccine to help reduce the growing number of people dying in the world due to COVID-19 was the main concern [153]. In this line, all kinds of resources and tools, both personal and financial, were made available to the pharmaceutical laboratories with the aim of obtaining a vaccine that would allow the whole world to return to normality, to resume contacts and, above all, to resume the economy and trade [154].

In this race towards a vaccine, several countries were working very fast, as the consequence of not doing so was an uncontrolled increase in the number of deaths [155]. However, there was a feeling in the general population that it was not possible to start a study on the disease and produce an effective vaccine in such a short time [156]. This perception was supported by fake news from groups of people who continue to argue that there is no vaccine for COVID-19 [157].

Once again, the lack of information among the general population, the lack of knowledge of the processes involved in creating a vaccine, and previous experience indicating that the average time to create a vaccine is at least ten years seriously damaged people’s perceptions of the arrival of a solution. In addition, once vaccination could be implemented in the weakest population, the elderly, the appearance of some side effects allowed these people who doubted the efficacy of the vaccine to reinforce these thoughts [128,136].

Another factor to consider, which has damaged the positive view of an effective remedy for the disease, is the appearance of some “effective treatments” for the cure of this virus through the networks and even by some of the world’s leading political figures [158]. Some of these treatments were directly related to conspiracy theories, such as the use of chlorine dioxide, and were based on the influence of the pharmaceutical industry, aiming to not support its use so that there would be a decrease in the sale of their medications [159,160].

All of these factors, combined with the fact that a debate was opened on social media from the beginning of the vaccine development process, have led to the spread of false information and misinformation [145]. These data that were being presented served to seriously alarm people who had an unrealistic perception of the disease and, in general, people who were confused by the events of the last year [130]. Once the vaccination process started, this whole situation made many people reluctant to be vaccinated because of various falsely propagated fears [161]. This has made it difficult to start the vaccination process in all countries, until, as the months have passed and the number of infections and deaths has fallen drastically, the population has become more aware of the benefits of vaccination [162].

An increasing number of people around the world have now been vaccinated to protect themselves and their families and loved ones from the disease [163]. However, there are still several people who do not want to be vaccinated and who remain a risk to themselves and others [164]. These people remain a risk to society, given the still significant presence of a virus that circulates globally and is sometimes lethal [165,166].

## 4. Study Limitations and Future Research

The main limitation of this review was the small quantity of empirical studies on this topic. It will be interesting to study this issue with more empirical evidence to build knowledge based on empirical evidence and to develop a meta-analysis. Future research should consider the evolution of misinformation and disinformation as the vaccination process evolves. 

## 5. Practical Statements

To combat misinformation and disinformation during a pandemic, institutions should bear in mind the situation of citizens (high stress and uncertainty) and the lack of resources to fight against it. The approach to fight misinformation and disinformation should be well-structured, prepared, and advanced, with previous preparation, social listening, understanding, and engagement. Transparency must be fostered from the institutions and centralized in clearly specified official communication channels. At the same time, institutions need to consider the necessity of the population to keep being informed through social media and public personalities. A multifactorial approach with the collaboration of social media companies and trustworthy public personalities should be enhanced to empower the correctness of valid information while stopping misleading information.

## 6. Conclusions

A pandemic usually comes along with a change in quality and perception of life because of the context of high uncertainty. It may influence people’s behaviors in different ways, such as skepticism, confusion, general malaise, or stress, which may foster their need to be informed. Technology can help misinformation to go viral and damage trust in scientists. Cybercriminals may find a new approach to steal people’s private data or to sale fake sanitary products and masks. The appearance of a pandemic generates uncertainty about what is true or false due to the lack of scientific evidence and the rapid growth of social media data. The vaccination process also plays an important role in a pandemic and must address similar challenges of untrust, uncertainty, and fear. The institutions should take seriously into account that a great part of the success of the intervention to combat a pandemic has a relationship with the power to stop misinformation and disinformation processes. The response should be well-structured and be addressed from different key points: at central level and community level, with official and clearly centralized communication channels. The approach should be multifactorial and enhanced from the collaboration of social media companies to stop misleading information, and trustworthy people both working and not working in the health care systems to boost the power of a message based on scientific evidence. 

## Figures and Tables

**Table 1 ijerph-19-05321-t001:** Social media platforms and their characteristics.

Social Media Platform	Start Year	Number of Monthly Active Users Worldwide in 2021 [117]	Characteristics	Dis/Misinformation Spread
Facebook	2004	2500 thousands of millions	Platform that allows users to upload, share and like various images, videos, live videos, stories, and specific pages.	It has deleted more than 20 million posts on its main social network and photo-sharing app Instagram for violating COVID-19 misinformation rules since the start of the pandemic. Facebook seeks to address criticism that its platforms have been used to spread fear about vaccines and misleading information about coronavirus. The company implemented new policies against COVID-19 misinformation, including a ban on repeat offenders spreading falsehoods and directing users to a central COVID-19 clearinghouse [118]. However, US President Joe Biden warned in July 2021 that the spread of misinformation about COVID-19 on social media is “killing people”, when questioned about the alleged role of “platforms like Facebook” in spreading falsehoods about vaccines and the pandemic [119].
YouTube	2005	2291 thousands of millions	Video sharing platform that allows users to upload, bookmark, and share videos.	YouTube said it removed 130,000 videos from its platform in 2020 and 2021 when it implemented a ban on content that spreads misinformation about COVID vaccines. Policy includes termination of antivaccine influencer accounts [119].
WhatsApp	2009	2000 thousands of millions	Instant messaging application for smartphones, in which messages are sent and received via the Internet, as well as images, videos, audio, audio recordings (voice notes), and calls and video calls with several participants at the same time, among other functions.	The ease of dissemination of messages from this social network and the relative anonymity that it provides to the first replicator of a chain message allows the sending of false and incomplete messages. In a study carried out in Zimbabwe countering misinformation through WhatsApp, it was found that potentially harmful behavior that does not comply with blocking guidelines decreased by 30 percentage points. The results show that social media posts from trusted sources can have substantially large effects not only on people’s knowledge but ultimately on related behavior [120]. Brazilian researchers questioned the role in the spread of the COVID-19 outbreak in their country and the disinformation through WhatsApp of President Bolsonaro’s narrative to minimize the impact of the disease in Brazil [121]
Instagram	2010	1000 thousand of millions	Image-sharing platform that allows users to upload, share, and like images and short videos.	An analysis of pages with hashtags frequently used by antivaccine conspirators found that general mistrust of vaccines was the most common, including the idea that the government and/or the media have fabricated or concealed information related to COVID-19. Conspiracy theories were the second most prevalent topic among the publications. In general, COVID-19 was frequently presented in association with beliefs that question authority [122]. A March 2021 report found that Instagram recommendations in the UK pushed users toward COVID disinformation, antivaccine content, and antisemitic material during the peak of the pandemic. Misinformation was most frequently displayed to new users who followed a combination of accounts on the platform that included leading personalities in the fight against vaccination or wellness influencers who disdained the efficacy of vaccines [123].
Twitter	2006	3295 thousands of millions	Platform that allows you to share short messages that can be accompanied by images.	Twitter introduced a feature in August 2021 that allows users to report misinformation they find on the platform, flagging it to the company as “misleading.” An exploratory study of my information found that false claims spread faster than partially false ones. Compared to a background corpus of COVID-19 tweets, misinformation tweets are more often concerned with discrediting other information on social media [124]. In August 2021, Twitter suspended 229 accounts and removed 5579 accounts that violated the antimisinformation policy of COVID-19 [125].

**Table 2 ijerph-19-05321-t002:** Stages of intervention for management to combat misinformation.

Stage	Characteristics
Preparación	It involves developing a personalized strategy for the public receiving the information, an evaluation of the information ecosystem, and the creation of the team with the right personnel.
Social listening	A social listening system can help optimize the detection of signs of misinformation and identification of emergencies or concerns from community members. The development of a social listening system must be guided by triangulation between the various tools available and the mapping of the information ecosystem, in particular, of the channels in which information related to vaccines is disseminated and discussed. Teams must ensure that they are equipped with the necessary skills to use these tools and make sense of the data to deliver actionable insights.
Understanding	Analyzing the potential impact of misinformation in a structured way helps classify rumors and identify rumors that require a response. It can be challenging to determine conclusively if something is true. The process requires research to obtain as much information as possible: verify the real origin of the information, the date of creation of the content, and the motivation for creating the content.
Engagement	Make sure that vaccine promotional content is more attractive (sticky) than misinformation. This can be achieved by capturing attention with high impact and visual media, presenting information clearly and continuously. Show the vaccination experience as positive avoiding the presentation of the act of vaccinating and crying children. Present stories of successful vaccination experiences.

Note. Source: Adapted from Unicef. Vaccine misinformation management field guide: Guidance for addressing a global infodemic and fostering demand for immunization (2020).

## Data Availability

Not applicable.

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
