# Peer review of "Mis–Dis Information in COVID-19 Health Crisis: A Narrative Review"

_ijerph, 2022, doi:10.3390/ijerph19095321_

Round 1

Reviewer 1 Report

Thank you very much for your work.

This review article analyzes mis/disinformation during the COVID-19 pandemic crisis as well as in the COVID-19 vaccination process. The barrage of mis/disinformation impacts health behavior and in the case of the current pandemic, can threaten the lives of individuals throughout the world. It is a timely topic and it is worth further exploration during the COVID-19 situation. The abstract shows a clear structure. The work presented is generally interesting and well explained. It summarizes the data from many relevant studies.

For the full paper, I would like to know more details about the following aspects.

  1. Authors present main findings in the section of “practical statements” and “conclusion”. From my viewpoint, it's better not to write it as a list.
  2. In the conclusion part, the author stated that “Transparency must be fostered from the institutions and centralized in clearly specified official communication channels”. The author also mentioned the measure taken by "social media companies to stop misleading information ". However, social media outlets have reportedly made efforts to limit false information, yet untruths related to COVID-19 persist online. In other words, we learn nothing new about the transmission mechanism of mis/dis information. Why can false information spread effectively and reach more people than true information? The discussion section seems to lack depth. It seems that further discussion is needed to answer this question. There have been many studies that have discussed how to combat mis/disinformation. The author can include these studies in this review article. In any case, this comment is only an observation that has arisen in my reading.

In summary, this article is timely, important and therefore suitable for the IJERPH readership.

Reviewer 2 Report

This is an interesting and important research, concerning current phenomena of COVID-19-related mis- and disinformation. The research method was selected and applied correctly. The conclusions from the narrative review is an important contribution for both future studies and practice related to information dissemination, monitoring or management. 

I suggest a few minor revisions to this text. 

First, in line 138 the authors suggest poor availability of the research literature on mis- and disinformation concerning the pandemic. However, the most simple search in Google Scholar reveals quite a lot such studies, so please revise your opinion in this matter.

Second, there is a list of main conclusions at the end of the text. However, neither limitations of the study no future research are mentioned or suggested. This would be beneficial for the article and the potential readers.

Author Response

Thank you for your comments. Please find attachment.
